# Targeting the vascular-specific phosphatase PTPRB protects against retinal ganglion cell loss in a pre-clinical model of glaucoma

Benjamin R Thomson[1,2], Isabel A Carota[1,2], Tomokazu Souma[1,2†], Saily Soman[1,2], Dietmar Vestweber[3], Susan E Quaggin[1,2]*

[1]Feinberg Cardiovascular and Renal Research Institute, Northwestern University Feinberg School of Medicine, Chicago, United States; [2]Division of Nephrology and Hypertension, Northwestern University Feinberg School of Medicine, Chicago, United States; [3]Max Planck Institute for Molecular Biomedicine, Münster, Germany

**Abstract** Elevated intraocular pressure (IOP) due to insufficient aqueous humor outflow through the trabecular meshwork and Schlemm's canal (SC) is the most important risk factor for glaucoma, a leading cause of blindness worldwide. We previously reported loss of function mutations in the receptor tyrosine kinase TEK or its ligand ANGPT1 cause primary congenital glaucoma in humans and mice due to failure of SC development. Here, we describe a novel approach to enhance canal formation in these animals by deleting a single allele of the gene encoding the phosphatase PTPRB during development. Compared to *Tek* haploinsufficient mice, which exhibit elevated IOP and loss of retinal ganglion cells, $Tek^{+/-};Ptprb^{+/-}$ mice have elevated TEK phosphorylation, which allows normal SC development and prevents ocular hypertension and RGC loss. These studies provide evidence that PTPRB is an important regulator of TEK signaling in the aqueous humor outflow pathway and identify a new therapeutic target for treatment of glaucoma.

*For correspondence:
quaggin@northwestern.edu

Present address: †Division of Nephrology, Duke University School of Medicine, Durham, United States

## Introduction

A leading cause of blindness worldwide, glaucoma is a devastating disease with no cure. Elevated intraocular pressure (IOP) caused by defects in the aqueous humor outflow (AHO) pathway is the most important risk factor for disease progression and vision loss (*Coleman and Miglior, 2008*; *Becker, 1961*). Indeed, IOP reduction is currently the only therapeutic intervention proven to slow glaucoma progression in patients. In both humans and mice, the majority of AHO occurs through the conventional route (*Johnson et al., 2017*; *Toris et al., 1999*; *Toris et al., 2000*; *Millar et al., 2011*; *Millar et al., 2015*), comprised of the trabecular meshwork (TM) and the large, lymphatic-like Schlemm's canal (SC) located in the iridocorneal angle (*Johnson et al., 2017*; *Park et al., 2014*; *Aspelund et al., 2014*; *Kizhatil et al., 2014*). Aqueous humor from the anterior chamber enters SC through the TM and is drained through a series of collector channels into the episcleral veins and systemic circulation. Recent studies have identified the importance of endothelial signaling molecules in development and maintenance of SC and the conventional outflow pathway, and critical roles have been described for the endothelial transcription factor PROX1 (*Park et al., 2014*) as well as the VEGFR2/3 (*Kizhatil et al., 2014*; *Aspelund et al., 2014*) and TEK (*Thomson et al., 2014*; *Kim et al., 2017*; *Souma et al., 2016*; *Thomson et al., 2017*) receptor tyrosine kinase signaling pathways.

TEK (Tunica interna endothelial cell kinase, also known as TIE2) is the tyrosine kinase receptor for the angiopoietin (Angpt) ligands ANGPT1, ANGPT2 and ANGPT4 (*Saharinen et al., 2017*). The

Angpt-TEK pathway is essential for SC development and maintenance, and loss of function mutations in *TEK* or the gene encoding its primary ligand *ANGPT1* have been identified in patients with primary congenital glaucoma, a severe form of glaucoma characterized by early/childhood onset, buphthalmos and optic neuropathy (*Souma et al., 2016*; *Thomson et al., 2017*; *Kabra et al., 2017*). Furthermore, recent genome-wide association studies have identified risk variants linked to the TEK signaling pathway in adults with elevated IOP and open angle glaucoma, the most common form of glaucoma worldwide (*Khawaja et al., 2018*; *Gao et al., 2018*; *MacGregor et al., 2018*).

In mice, post-natal *Tek* deletion leads to complete failure of SC development and rapidly progressing glaucoma (*Thomson et al., 2014*; *Kim et al., 2017*; *Souma et al., 2016*). A similar disease is observed in mice lacking the TEK ligand ANGPT1, confirming that ANGPT-TEK signaling is essential for canal development (*Thomson et al., 2017*). Importantly, while *Tek* knockout mice exhibit complete loss of SC, a hypomorphic canal characterized by focal narrowing and convolutions is observed in haploinsufficient animals (*Tek*$^{+/-}$ mice) (*Souma et al., 2016*). This hypomorphic SC is associated with moderate IOP elevation, indicating a clear dose-dependent effect of TEK signaling in development and function of the aqueous outflow pathway and suggesting that TEK activation using genetic or pharmacological approaches might provide novel treatments for patients with high-pressure glaucoma. Indeed, the clear dose-dependent relationship between ANGPT-TEK signaling and severity of disease presentation in rodent models supports the argument that therapeutic modulation of this pathway will be efficacious in patients (*Plenge et al., 2013*).

Activation of the TEK receptor has been achieved in vitro and in vivo either by increasing activity of endogenous ANGPT ligands, providing ANGPT recombinant proteins (*Kim et al., 2005*; *Souma et al., 2018*), or by suppression of the phosphatase PTPRB (also known as the Vascular Endothelial Protein Tyrosine Phosphatase, VE-PTP) (*Winderlich et al., 2009*; *Shen et al., 2014*; *Carota et al., 2019*), which strongly dephosphorylates TEK (*Souma et al., 2018*; *Fachinger et al., 1999*). PTPRB inhibition results in ligand-independent increased TEK phosphorylation at all phosphorylated tyrosine residues, and leads to a dramatic increase in downstream signaling (*Souma et al., 2018*; *Carota et al., 2019*). Here, we show that developmental deletion of a single *Ptprb* allele in mice is sufficient to reduce PTPRB expression and leads to increased TEK activation in vivo. Furthermore, in *Tek*$^{+/-}$ haploinsufficient mice, this increased TEK activation is sufficient for normal SC development, preventing both ocular hypertension and retinal ganglion cell (RGC) loss.

## Results

To increase the level of TEK phosphorylation in vivo, we utilized a *Ptprb*$^{NLS-LacZ}$ knock-in reporter allele (*Bäumer et al., 2006*) to delete a single allele of the *Ptprb* gene. This construct incorporates a β-Galactosidase cDNA tagged with a nuclear localization signal in place of the first exon of *Ptprb*, preventing production of PTPRB protein. As previously described, heterozygous *Ptprb*$^{NLS-LacZ/WT}$ mice are born normally (*Bäumer et al., 2006*), although expression of PTPRB was reduced by approximately 50% (*Figure 1A*, uncropped images presented as *Figure 1—figure supplement 1*). Likewise, *Tek* heterozygosity resulted in approximately 50% reduction in TEK protein detected in lung lysate (*Figure 1A*). Reductions in phosphatase abundance had a direct effect on TEK activation and *Ptprb*$^{NLS-LacZ/WT}$ mice showed approximately a 118% increase in phosphorylated TEK when measured in lung tissue using an immunoprecipitation assay (*Figure 1B*, uncropped images presented as *Figure 1—figure supplement 2*), confirming our hypothesis that changes in PTPRB expression would have a direct impact on TEK phosphorylation.

The finding that *Ptprb*$^{NLS-LacZ/WT}$ haploinsufficient mice exhibited markedly increased TEK phosphorylation suggested that these animals would provide a powerful tool to study the effect of PTPRB blockade on SC development in the context of reduced TEK signaling. Therefore, we crossed mice carrying the *Ptprb*$^{NLS-LacZ}$ allele with a previously-described mouse model of *Tek* heterozygosity, creating a constitutive model of *Tek*$^{+/-}$;*Ptprb*$^{NLS-LacZ/WT}$ double haploinsufficiency. Adult *Tek* haploinsufficient mice have been reported to exhibit a hypomorphic SC insufficient for normal AHO, leading to moderate IOP elevation (*Souma et al., 2016*). To test our hypothesis that ~50% reduction of PTPRB activity might prevent the glaucoma phenotype in *Tek*$^{+/-}$ mice without the need to increase ligand availability, *Tek*$^{+/-}$;*Ptprb*$^{NLS-LacZ/WT}$ double heterozygous mice were generated, and enucleated eyes were collected. After fixation, eyes were prepared for whole-mount immunostaining and visualized using confocal microscopy. Consistent with previous findings (*Souma et al., 2016*), analysis of CD31-

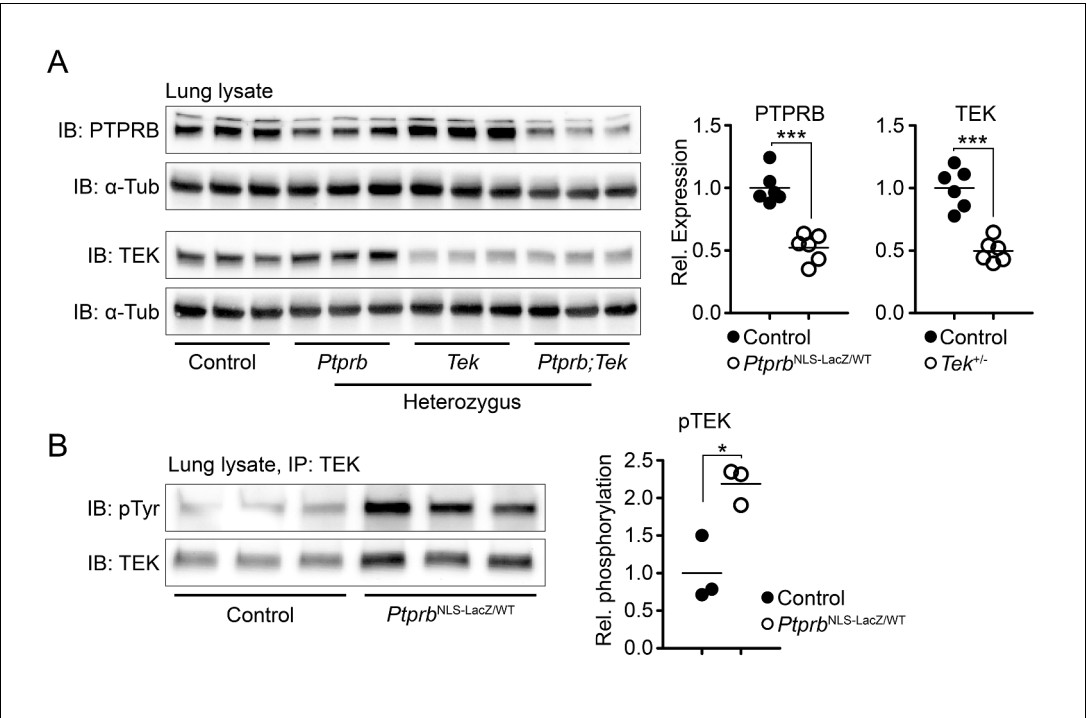

**Figure 1.** Deletion of one *Ptprb* allele leads to increased TEK phosphorylation. (**A**) Western blot of lung lysate from P5 Control, *Tek*[+/-], *Ptprb*[NLS-LacZ/WT] and *Tek*[+/-];*Ptprb*[NLS-LacZ/WT] mice revealed a 50% reduction in PTPRB expression in *Ptprb*[NLS-LacZ/WT] (heterozygous) and *Tek*[+/-];*Ptprb*[NLS-LacZ/WT] mice. Likewise, TEK expression was reduced approximately 50% in *Tek*[+/-] and *Tek*[+/-];*Ptprb*[NLS-LacZ/WT] mice. (**B**) Immunoprecipitation of lung lysates from adult control and *Ptprb*[NLS-LacZ/WT] using anti-TEK antibody followed by western blotting with anti-phospho tyrosine antibody revealed a marked elevation of TEK phosphorylation in *Ptprb*[NLS-LacZ/WT] animals compared to littermate controls. Horizontal lines indicate population means. *$p<0.05$, ***$p<0.001$ as determined by Student's t-test.

The online version of this article includes the following figure supplement(s) for figure 1:

**Figure supplement 1.** Uncropped images corresponding to the western blots presented in *Figure 1A*.
**Figure supplement 2.** Uncropped images corresponding to the western blots presented in *Figure 1B*.

---

positive SC area revealed a hypomorphic canal phenotype in adult *Tek* heterozygous mice when compared to littermate controls (Control: 29,974 ± 2145, *Tek*[+/-]: 17,457 ± 1040 μm$^2$/20x field; *Figure 2A*). This phenotype was abrogated by deletion of a single *Ptprb* allele in *Tek*[+/-];*Ptprb*[NLS-LacZ/WT] animals, which exhibited normal SC area (*Tek*[+/-];*Ptprb*[NLS-LacZ/WT]: 23,848 ± 1574 μm$^2$/20x field). Importantly, despite elevated TEK activation, *Ptprb*[NLS-LacZ/WT] animals displayed a normal SC (*Ptprb*[NLS-LacZ/WT]: 28,175 ± 1668 μm$^2$/20x field), suggesting reduction of PTPRB function is well tolerated during SC development in the absence of other mutations.

Developmental haploinsufficiency of *Tek* and *Ptprb* had clear effects on adult SC area in our model. To determine if this phenotype was due to altered proliferation of SC endothelial cells (ECs), we next analyzed eyes collected at P5 when SC development is underway. In *Tek*[+/-] eyes, we observed a marked reduction in the proportion of PROX1+ SC ECs which were also positive for Ki-67 (Control: 0.302 ± 0.022, *Tek*[+/-]: 0.214 ± 0.016), suggesting that the hypomorphic canal phenotype in these animals may be due to reduced proliferation during canal development (*Figure 2B*, quantified in **C**). Strikingly, normal proliferation of PROX1+ SC cells was observed in *Tek*[+/-];*Ptprb*[NLS-LacZ/WT] mice (*Tek*[+/-];*Ptprb*[NLS-LacZ/WT]: 0.345 ± 0.039), providing a potential mechanism for the observed effects on canal area in adulthood. A similar effect was observed on PROX1 expression, which was dramatically decreased in the SC of *Tek*[+/-] eyes at P5 and appeared normal in the eyes of *Tek*[+/-];*Ptprb*[NLS-LacZ/WT] littermates (Norm. AFU: Control: 1.0 ± 0.12, *Tek*[+/-]: 0.49 ± 0.07, *Tek*[+/-];*Ptprb*[NLS-LacZ/WT]: 0.82 ± 0.065, *Figure 2D*).

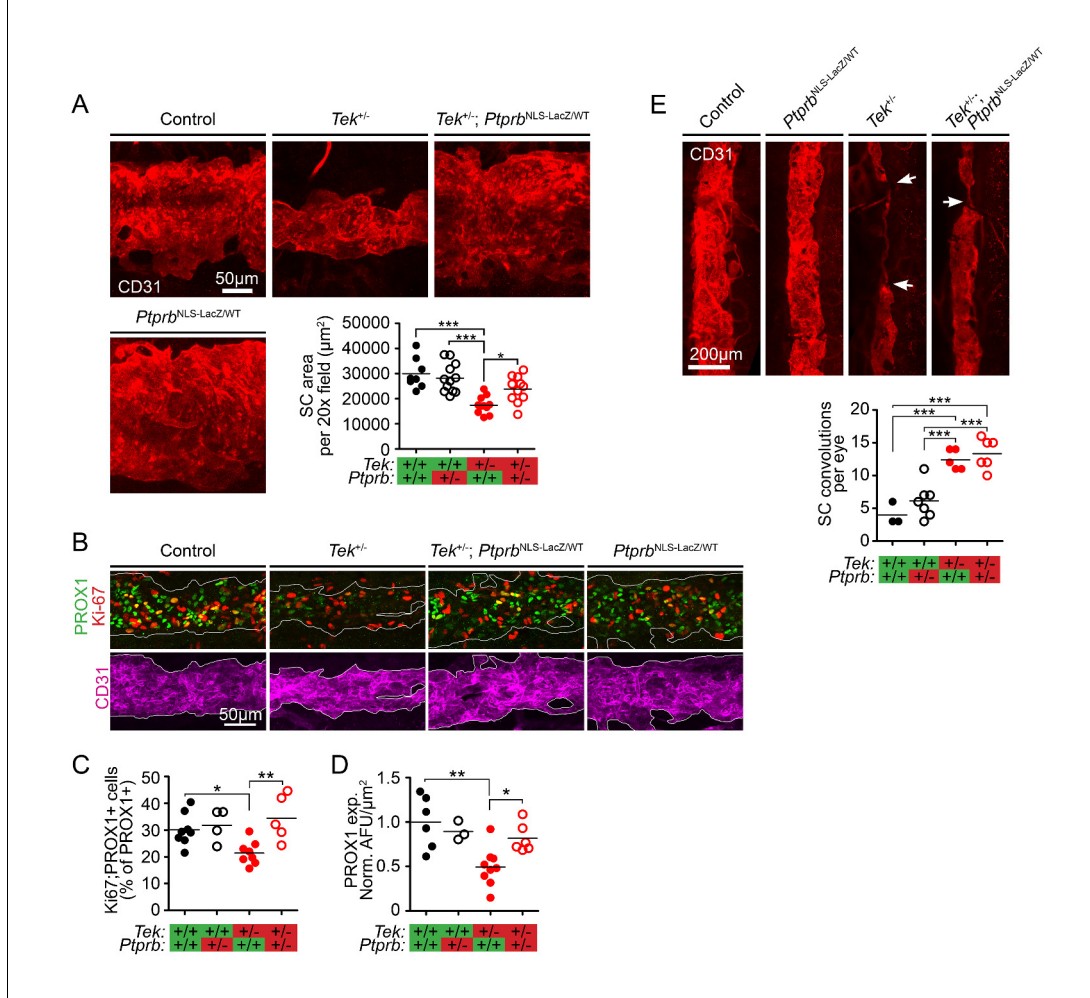

**Figure 2.** TEK signaling has a dose-dependent effect on Schlemm's canal (SC) area and development. (**A**) Confocal microscopy of whole mount eyes revealed reduced CD31+ SC area in adult $Tek^{+/-}$ haploinsufficient mice. This phenotype was blunted in $Tek^{+/-};Ptprb^{NLS-LacZ/WT}$ double heterozygous animals, confirming the importance of TEK activation in canal development. $Ptprb^{NLS-LacZ/WT}$ heterozygous controls had normal SC area (n = 8 WT, 12 $Ptprb^{NLS-LacZ/WT}$, 11 $Tek^{+/-}$ and 11 $Tek^{+/-};Ptprb^{NLS-LacZ/WT}$ mice). 20x fields shown represent an area of 65,536 $\mu m^2$. Images were captured as 10-frame Z stacks with a step size of 1.67 $\mu$m and a pinhole of 1.2 Airy units, and are shown as maximum intensity projections. (**B**, quantified in **C**) At postnatal day 5 (P5), confocal microscopy of the developing SC in eye whole mounts revealed reduced numbers of proliferating Ki-67-positive SC ECs (Ki67-PROX1 double positive cells) in $Tek$ haploinsufficient animals compared to littermate WT or $Ptprb^{NLS-LacZ/WT}$ controls. Normal proliferation was observed in $Tek^{+/-};Ptprb^{NLS-LacZ/WT}$ animals. (**D**) Compared to control and $Ptprb^{NLS-LacZ/WT}$ mice, PROX1 expression was reduced in $Tek^{+/-}$ littermate eyes. Expression was normal in $Tek^{+/-};Ptprb^{NLS-LacZ/WT}$ double heterozygotes. n = 8 (WT), 4 ($Ptprb^{NLS-LacZ/WT}$), 8 ($Tek^{+/-}$) and 5 ($Tek^{+/-};Ptprb^{NLS-LacZ/WT}$) Shown are maximum intensity projections from 8-frame confocal Z stacks captured using a 20x objective, step size of 1 $\mu$m and pinhole of 1.2 Airy units. Norm. AFU: Normalized, background subtracted arbitrary fluorescence units. (**E**) Compared to control and $Ptprb^{NLS-LacZ/WT}$ littermates, confocal analysis of adult SC revealed a marked increase in the number of focal convolutions and narrowings in the eyes of $Tek^{+/-}$ and $Tek^{+/-};Ptprb^{NLS-LacZ/WT}$ mice. N = 4 (WT), 7 ($Ptprb^{NLS-LacZ/WT}$), 5 ($Tek^{+/-}$) and 6 ($Tek^{+/-};Ptprb^{NLS-LacZ/WT}$). Horizontal lines indicate population means. *p<0.05, **p<0.01, ***p<0.001 as determined by 1-way ANOVA followed by Bonferroni's correction.

In addition to decreased area, $Tek^{+/-}$ SCs are characterized by focal thinning and convolutions which are not present in WT littermates (*Souma et al., 2016*) (*Figure 2E*). Interestingly, although $Ptprb$ haploinsufficiency resulted in increased proliferation and expanded canal area in $Tek^{+/-};$ $Ptprb^{NLS-LacZ/WT}$ animals, these focal defects were still observed—suggesting that directional signaling may play a role in proper canal formation which cannot be recapitulated by nonspecific TEK activation.

We next examined the effect of altered SC area in $Tek^{+/-}$ and $Tek^{+/-};Ptprb^{NLS-LacZ/WT}$ mice on aqueous humor homeostasis. $Tek$ heterozygous mice on an outbred background were found to have

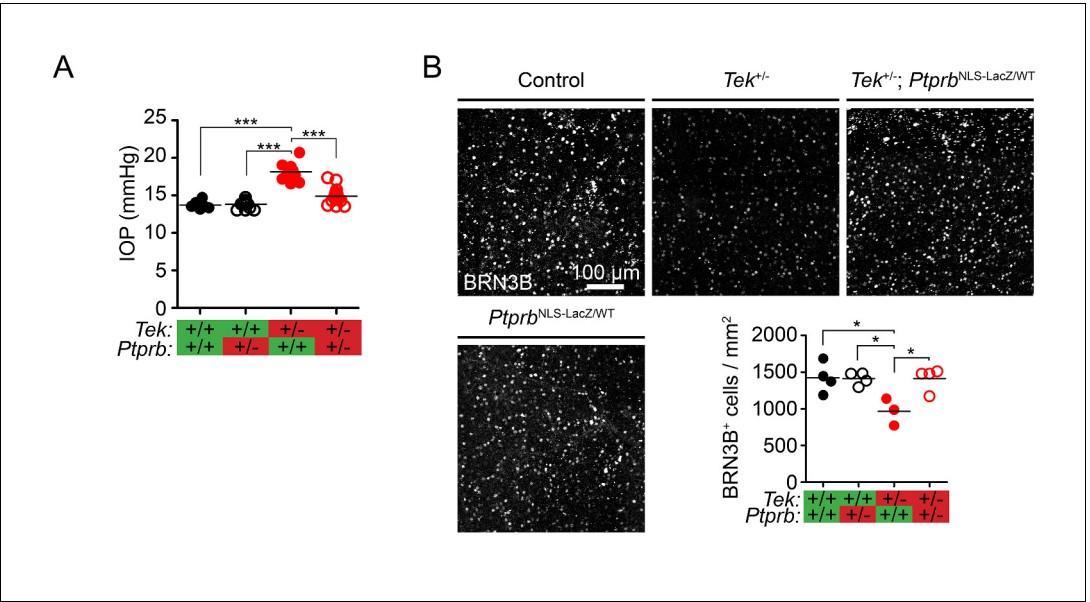

**Figure 3.** *Ptprb* heterozygosity prevents ocular hypertension and RGC loss in *Tek* haploinsufficient mice. (**A**) Elevated intraocular pressure (IOP) was observed in *Tek*$^{+/-}$ haploinsufficient mice at 30 weeks of age when measured by rebound tonography. As in *Figure 2*, this phenotype was prevented in *Tek*$^{+/-}$;*Ptprb*$^{NLS-LacZ/WT}$ double heterozygous animals, confirming the importance of TEK activation in IOP homeostasis (n = 6 WT, 14 *Ptprb*$^{NLS-LacZ/WT}$, 12 *Tek*$^{+/-}$ and 14 *Tek*$^{+/-}$;*Ptprb*$^{NLS-LacZ/WT}$ mice). (**B**) BRN3B staining in retinal flat-mounts from a second group of mice revealed loss of retinal ganglion cells by 19 weeks in *Tek*$^{+/-}$ mice. Littermate *Tek*$^{+/-}$;*Ptprb*$^{NLS-LacZ/WT}$ animals were protected, correlating with the reduced IOP observed (n = 4 WT, 4 *Ptprb*$^{NLS-LacZ/WT}$, 3 *Tek*$^{+/-}$ and 4 *Tek*$^{+/-}$;*Ptprb*$^{NLS-LacZ/WT}$ mice). Horizontal lines indicate population means. *p<0.05, ***p<0.001 as determined by 1-way ANOVA followed by Bonferroni's correction.

The online version of this article includes the following figure supplement(s) for figure 3:

**Figure supplement 1.** Retinal vasculature was normal in *Tek* and *Ptprb* haploinsufficient mice.

---

elevated IOP at 30 weeks of age (*Figure 3A*, Control: 13.7 ± 0.23, *Tek*$^{+/-}$ 18.15±0.33 mmHg). Consistent with observations of SC morphology, while *Ptprb*$^{NLS-LacZ/WT}$ mice had normal IOP (13.81 ± 0.82 mmHg), incorporation of this allele into the *Tek*$^{+/-}$ model was beneficial and blunted the ocular hypertension associated with *Tek* haploinsufficiency, despite the presence of focal morphological defects (*Tek*$^{+/-}$;*Ptprb*$^{NLS-LacZ/WT}$ IOP: 14.92 ± 0.31 mmHg). To confirm the impact of the *Ptprb*$^{NLS-LacZ}$-induced IOP reduction on the retina, we counted BRN3B positive ganglion cells in an additional cohort of mice at 19 weeks of age (*Figure 3B*). While *Tek* heterozygous mice showed a marked reduction in RGCs, this loss did not occur in *Tek*$^{+/-}$;*Ptprb*$^{NLS-LacZ/WT}$ mice.

TEK signaling is critical for retinal angiogenesis, and to exclude the possibility that vascular defects and subsequent ischemia were responsible for RGC loss in *Tek*$^{+/-}$ mice, we examined vascular morphology in our animals. We first examined retinas collected at P5, when development of the superficial vasculature is ongoing. At this timepoint, CD31 staining revealed normal progress of the angiogenic sprouting front in *Tek*$^{+/-}$, *Ptprb*$^{NLS-LacZ/WT}$, and *Tek*$^{+/-}$;*Ptprb*$^{NLS-LacZ/WT}$ mice when compared to littermate controls (*Figure 3—figure supplement 1A*). At P20, when development of the mature retinal vasculature is complete, we observed normal patterning in all three retinal vascular layers in mice of all genotypes (*Figure 3—figure supplement 1B*). By this timepoint, hyaloid vessels were not observed in animals of any genotype. Taken together, these data suggest that blunted IOP elevation in *Tek*$^{+/-}$;*Ptprb*$^{NLS-LacZ/WT}$ animals likely had a direct impact on glaucoma pathogenesis in our model and that retinal ischemia is unlikely to account for the RGC loss observed in Tek haploinsufficient mice.

## Discussion

Ocular hypertension is the single most important risk factor for glaucoma in patients, and reducing IOP is the only therapeutic intervention proven to slow disease progression. Although aqueous humor flow through the conventional (trabecular) outflow pathway represents the majority of AHO, drugs targeting this outflow pathway were not available until the recent introduction of the Rho kinase inhibitors ripasudil and netarsudil. However, while these molecules provide exciting new treatment options, clinical trials have not shown them to be superior to current standard of care therapies for lowering IOP (*Tanna and Johnson, 2018*), underscoring the need for additional drug targets and treatment options.

Angpt-TEK signaling is essential for development (*Thomson et al., 2014*; *Souma et al., 2016*; *Thomson et al., 2017*) and maintenance (*Kim et al., 2017*) of SC, an essential component of the conventional AHO pathway. Dysregulation of Angpt-TEK signaling results in hypomorphic or failed SC formation, elevated IOP and glaucoma in mice and humans. Intriguingly, recent GWAS studies of subjects with ocular hypertension have identified common variants in members of the Angpt-TEK signaling pathway (*Khawaja et al., 2018*; *Gao et al., 2018*; *MacGregor et al., 2018*). These GWAS findings suggest that in addition to the complete loss-of-function variants previously described in PCG, less impactful variants may play a role in more common forms of adult-onset glaucoma. These reports, combined with the strong dose dependent response of SC development and function to TEK signaling observed in the mouse, suggest that the Angpt-TEK signaling axis is a sensitive and important regulator of SC and AHO and may provide a valuable IOP-lowering therapeutic target for treatment of glaucoma patients. Indeed, a recent study has demonstrated the effectiveness of targeting the Angpt-TEK signaling pathway in a mouse model of glaucoma using an antibody which increases the signaling ability of the context-dependent, weak TEK agonist ANGPT2 by clustering it into higher-order multimers (*Kim et al., 2017*; *Park et al., 2016*). ANGPT2-clustering antibody treatment was found to increase TEK phosphorylation in aged mice and lower IOP in an injury-induced model of ocular hypertension (*Kim et al., 2017*).

Here, we demonstrate an alternative approach targeting the endothelial phosphatase PTPRB, which acts on TEK (*Souma et al., 2018*; *Winderlich et al., 2009*; *Bäumer et al., 2006*) as well as VE-Cadherin, VEGFR2 and FGD5 (*Nawroth et al., 2002*; *Broermann et al., 2011*; *Mellberg et al., 2009*; *Hayashi et al., 2013*; *Braun et al., 2019*). Using a genetic approach to PTPRB ablation allowed us to achieve consistent reduction in protein levels, with deletion of a single *Ptprb* allele leading to approximately 50% reduction in protein expression. In *Tek*-WT lung tissue, this was sufficient to cause a marked elevation in TEK phosphorylation. PTPRB inhibition has been previously studied as a potential therapeutic intervention in vascular disease (*Shen et al., 2014*), and the PTPRB inhibitor AKB-9778 is currently undergoing clinical trials for treatment of diabetic macular edema, where it is well-tolerated by patients (*Campochiaro et al., 2016*). While mid-term results for the AKB-9778 TIME-2 trial failed to achieve the primary endpoint in diabetic macular edema, two important observations were reported from this clinical trial: the drug was well tolerated and safe in patients and IOP was lower in the treatment arm (*Aerpio Pharmaceuticals Inc, 2019*).

We have previously reported that defects in Angpt-TEK signaling cause a range of SC and IOP phenotypes corresponding to the degree of pathway dysfunction, ranging from complete absence of SC (*Tek* knockout or *Angpt1;Angpt2* double knockout mice) to less severely hypomorphic canal morphology (*Tek* haploinsufficiency) (*Thomson et al., 2014*; *Kim et al., 2017*; *Souma et al., 2016*; *Thomson et al., 2017*). Although the phenotype of *Tek* haploinsufficient mice is milder than that observed in PCG patients with heterozygous *TEK* loss-of-function mutations (*Souma et al., 2016*), we selected this model as *Tek* heterozygous mice are viable, fertile and exhibit a consistent SC phenotype without the need for Cre recombinase-mediated gene deletion. Furthermore, this mouse model has many features of glaucoma including ocular hypertension and loss of retinal ganglion cells which more faithfully recapitulate disease seen in patients with common forms of open angle glaucoma then the more severe model of total *Tek* deletion. Consistent with our previous findings, adult $Tek^{+/-}$ mice in this study displayed a hypomorphic SC phenotype due to decreased EC proliferation which was associated with elevated IOP and loss of BRN3B+ retinal ganglion cells. These defects were blunted by developmental *Ptprb* haploinsufficiency in $Tek^{+/-};Ptprb^{NLS-LacZ/WT}$ mice, likely due to increased basal TEK phosphorylation and resulting EC proliferation in $Ptprb^{NLS-LacZ/WT}$ animals. In addition, expression of PROX1, a key marker of mature SC endothelial cells, was dramatically

reduced in $Tek^{+/-}$ SCs at P5. Expression was normal in $Tek^{+/-};Ptprb^{NLS-LacZ/WT}$ eyes, consistent with previous reports from our lab and elsewhere suggesting that Angpt1-TEK signaling may be critical for maintenance of this crucial transcription factor in the SC endothelium (*Park et al., 2014*; *Kim et al., 2017*; *Thomson et al., 2017*; *Truong et al., 2014*). However, our data do not indicate whether PROX1 expression is directly downstream of TEK activation or is otherwise dependent on a mature SC endothelial phenotype.

While these results highlight the potential of PTPRB as a therapeutic target for patients with TEK-associated PCG, the recent association of Angpt-TEK variants with ocular hypertension and the findings of Kim et al using an ANGPT2 clustering antibody suggest that this pathway may also provide a valuable therapeutic target for adult patients with primary open angle glaucoma, a significantly larger group then TEK-associated PCG. Future studies using targeted gene deletion or therapies that inhibit PTPRB such as siRNA, antibodies or small molecule inhibitors will provide additional insights and pave the way for translation of these findings into the clinic.

# Materials and methods

## Key resources table

| Resource | Designation | Source or reference | Identifiers | Additional information |
|---|---|---|---|---|
| Genetic Reagent (*M. musculus*) | $Ptprb^{NLS-LacZ}$ | *Bäumer et al., 2006* | | Maintained on a mixed background |
| Genetic Reagent (*M. musculus*) | $Tek^{+/-}$ | *Thomson et al., 2017* | $Tek^{tm1.1Vlcg}$; MGI:5544795 | Maintained on a mixed background |
| Antibody | anti-PTPRB (Rabbit polyclonal) | *Nawroth et al., 2002* | | Western blot: 1:2000 |
| Antibody | Anti-TEK (Rabbit polyclonal) | Santa Cruz Biotech | sc-324 | Western blot 1:2500 |
| Antibody | anti-αTubulin (Mouse monoclonal) | Santa Cruz Biotech | sc-32293 | Western blot: 1:10,000 |
| Antibody | 4G10 Platinum anti-phosphotyrosine (Mouse monoclonal) | Millipore | 05–1050 | Western blot: 1:2000 |
| Antibody | anti-CD31 MEC13.3 (Rat monoclonal) | BD Biosciences | 55337 | IF: 1:100 |
| Antibody | anti-PROX1 (Goat polyclonal) | R and D Systems | AF2727 | IF: 1:200 |
| Antibody | anti-Ki-67 (Rabbit monoclonal) | ThermoFisher | MA5-14520 | IF: 1:200 |
| Antibody | anti-BRN3b (Goat polyclonal) | Santa Cruz Biotech | sc-6026 | IF: 1:1000 |
| Software, algorithm | ImageJ Fiji | *Schindelin et al., 2012* | Version 1.52 p | Used for all image analysis |
| Software, algorithm | Graphpad Prism | Graphpad.com | Version 5.0 | Used for statistical analysis and graph generation |
| Software, algorithm | Adobe Indesign | Adobe.com | Version 14.01 × 64 | Used for figure creation |

## Study approval

This study was performed in strict accordance with the recommendations in the Guide for the Care and Use of Laboratory Animals of the National Institutes of Health and the ARVO guidelines for care and use of vertebrate research subjects in eye research. All animal experiments were approved by the Animal Care Committee at the Center for Comparative Medicine of Northwestern University (Evanston, Illinois, USA).

## Generation and breeding of Tek[+/-];Ptprb[NLS-LacZ/WT] mice

Ptprb[NLS-LacZ/WT] mice have been previously described (*Winderlich et al., 2009*; *Bäumer et al., 2006*) and were a generous gift of Dr. Dietmar Vestweber (Max Planck Institute, Münster, Germany). Tek[+/-] mice were generated by crossing Tek[COIN] (*Thomson et al., 2014*; *Economides et al., 2013*) mice with Rosa26[rtTA];TetOnCre (*Belteki et al., 2005*) as previously described (*Souma et al., 2016*). After undergoing Cre-mediated gene deletion, animals were crossed with WT ICR mice to obtain Tek[+/-] animals which did not express the TetOnCre or Rosa26[rtTA] transgenes. Throughout the present study, animals were maintained on a mixed genetic background free of the retina degeneration mutations RD1 and RD8 and allowed unrestricted access to standard rodent chow (Harlan Teklad #7912; Envigo, Indianapolis IN) and water. As animals were maintained on a mixed background, littermate controls were used for all experiments and animals were included in the study on the basis of full litters (i.e. a control from one litter would not be included without their matching mutant littermates). To determine experimental group sizes, data from our previous studies of SC morphology (*Souma et al., 2016*; *Thomson et al., 2017*) were used to estimate required numbers. Breeding cages were then set up based on these estimates, and all resulting animals were included in the described studies.

## Western blot

Lung samples from Tek[+/-], Ptprb[NLS-LacZ/WT] and Tek[+/-], Ptprb[NLS-LacZ/WT] mice with control littermates were homogenized in RIPA buffer (50 mM TRIS, 150 mM NaCl, 1% IGEPAL CA-630, 0.5% Na Deoxycholate, 0.1% SDS, pH 7.5) containing protease and phosphatase inhibitor cocktails (Sigma). Samples were lysed (30 min at 4°C) and centrifuged (10 min at 14,000*g, 4°C) before the supernatant was used for western blot and immunoprecipitation. For western blot, 100 µg lysate was separated on a 4–15% Tris-glycine gel (Bio-Rad) and transferred to PVDF membranes using standard methods. Membranes were then cut horizontally at 75 kDa and blocked in (5% BSA in Tris buffered saline containing 0.05% Tween-20, pH 7.5) before incubating with appropriate primary and HRP-conjugated secondary antibodies (Jackson Immunoresearch). After washing, membranes were incubated in ECL substrate (SuperSignal West Pico PLUS, Thermo Fisher) and imaged on an iBright 1500 digital camera system (Thermo Fisher-Life Technologies). Quantification was performed using ImageJ Fiji software (*Schindelin et al., 2012*). Values obtained from quantifying total TEK or PTPRB were normalized to matching alpha-tubulin bands obtained from the lower (<75 kDA) region of the same membrane. Primary antibodies used: Rabbit anti-TEK (Santa Cruz Biotech #sc-324, 1:2500), mouse anti-alpha tubulin (Santa Cruz Biotech #sc-32293, 1:10,000), rabbit anti-PTPRB (*Nawroth et al., 2002*).

For Immunoprecipitation assays of TEK phosphorylation, 1 mg of protein lysate was incubated with 1 µg rabbit anti-TEK antibody (C-20, Santa Cruz) before antibody-protein complexes were captured using Protein-A conjugated Dynabeads (Invitrogen). Proteins were then eluted by boiling in 2x Laemmli sample buffer containing 100 mM DTT, loaded on a 4–15% Tris-glycine gel and separated by SDS-PAGE as described above. Phosphorylated tyrosine was detected using mouse anti-phospho-tyrosine antibody (4G10 Platinum, Millipore #05–1050, 1:2000) before membranes were stripped using a commercial stripping solution (Restore, Thermo Fisher # 21059) and re-probed using anti-TEK antibody as described above. Bands were imaged using a ChemiDock imaging system (Bio-Rad) and quantified as above. Relative TEK phosphorylation is reported in the manuscript as a normalized ratio of pTyr:Total TEK signals obtained from the same membrane. All western blot and immunoprecipitation experiments were performed at least twice. Values and statistics reported are derived from the data shown in the manuscript.

## Schlemm's canal immunostaining and imaging

Whole-mount imaging of SC was performed as described previously (*Thomson et al., 2017*; *Thomson and Quaggin, 2018*). Briefly, enucleated eye globes were immersion fixed (2% formaldehyde in 0.1M phosphate buffer pH 7.5, 12 hr at 4°C) before the lens and retina were removed and limbal flat mounts were prepared. Tissues were blocked (5% donkey serum, 2.5% bovine serum albumin in Tris buffered saline pH 7.5 containing 0.5% Triton X-100, overnight at 4°C) before incubating in appropriate primary and alexafluor-labled secondary antibodies (Thermo Fisher Scientific, Waltham, MA) diluted in additional blocking buffer. Antibodies used: Rat anti-mouse CD31 (Dilution

1:100. #55337, BD Biosciences, Franklin Lakes New Jersey), Rabbit anti-human Ki-67 (Dilution 1:200. #MA5-14520, ThermoFisher Scientific, Waltham, MA), goat anti-human PROX1 (Dilution 1:200. #AF2727, R and D Systems, Minneapolis, MN). After staining, tissues were washed (Tris buffered saline pH 7.5 containing 0.05% Tween-20) and mounted on microscope slides. Images were captured using a Nikon A1R confocal microscope at the Center for Advanced Microscopy at Northwestern University equipped with a 20x objective with a numerical aperture of 0.75. To measure SC area, 3 10-image Z stacks were captured using a step size of 1.67 µm and a pinhole of 1.2 Airy units at intervals around the circumference of SC. To prevent bias in image location selection, the episcleral vein was used as a landmark for the initial field and subsequent images were taken at 120° intervals. In the manuscript, images of SC are shown as maximum intensity projections of these 10-image Z stacks. Canal area was measured in each 65,536 $µm^2$ field using Fiji software (*Schindelin et al., 2012*) and an average value was obtained for each eye. Likewise, quantification of Ki-67 and PROX1 expression was performed in three imaging zones from each eye and the results were averaged to obtain the reported value for the animal. eight image confocal Z stacks were obtained with a step size of 1 µm and pinhole of 1.2. Ki-67 positive nuclei were counted manually using ImageJ Fiji software. PROX1 expression was reported as mean background-subtracted fluorescence per $µm^2$ CD31+ SC area as measured using ImageJ Fiji software. Quantification of convolutions and focal defects was performed from stitched images of the full SC circumference obtained using the objective described above with the pinhole set to 150 µm.

## Intraocular pressure measurement

IOP measurements were obtained from awake mice using a Tonolab rebound tonometer (iCare) as previously described (*Thomson et al., 2014*; *John et al., 1997*). Animals were restrained in a soft plastic cone, and average IOPs were recorded from 3 sets of 6 recordings performed by a blinded technician. Finding no difference between left and right eyes, we have reported all IOP measurements as single averaged values for each animal.

## Retinal ganglion cell quantification

Enucleated eyes from $Tek^{+/-};Ptprb^{NLS-LacZ/WT}$ mice with littermate $Tek^{+/-}$, $Ptprb^{NLS-LacZ/WT}$, and WT controls were collected at 19 weeks and fixed as above. Retinas were collected before blocking and staining as above. Antibodies used: Goat anti BRN3 (Dilution: 1:1000. Santa Cruz Biotechnology, sc-6026). Retinas were imaged in a standardized pattern as flat mounts using a Nikon A1R microscope as described above and BRN3+ ganglion cells were quantified by a blinded student.

## Statistical analysis

Statistical analysis was performed using Prism 5.0 software (Graphpad, La Jolla CA USA). Two-tailed Student's t-test or ANOVA followed by use of Bonferroni's method for multiple comparisons were used for statistical significance testing as appropriate. Throughout the text, values are reported as means ± standard error (SEM). In figures, plotted data points represent average values from individual animals with horizontal lines indicating the group mean. p-values<0.05 were considered significant and are indicated in figures using the following notation: *p<0.05, **p<0.01 and ***p<0.001.

## Acknowledgements

We are grateful to Megan Kelly and Veronica Ramirez for technical assistance. This study was funded by NIH R01 HL124120, R01 EY025799 (to SEQ). Imaging was performed at the Northwestern University Center for Advanced Microscopy supported by NCI CCSG P30 CA060553 awarded to the Robert H Lurie Comprehensive Cancer Center. We also acknowledge support from the George M O'Brien Kidney Core Grant P30 DK114857.

## Additional information

### Competing interests

Isabel A Carota: was employed by Eli Lilly and Company during the time of study completion and manuscript preparation. Dietmar Vestweber: is a scientific advisory board member of Aerpio Pharmaceuticals. Susan E Quaggin: has applied for patents related to therapeutic targeting of the ANGPT-TEK pathway in ocular hypertension and glaucoma and receives research support, owns stock in and is a director of Mannin Research. The other authors declare that no competing interests exist.

### Funding

| Funder | Grant reference number | Author |
| --- | --- | --- |
| National Institutes of Health | R01 HL124120 | Susan E Quaggin |
| National Institutes of Health | R01 EY025799 | Susan E Quaggin |
| National Institutes of Health | P30 DK114857 | Susan E Quaggin |

The funders had no role in study design, data collection and interpretation, or the decision to submit the work for publication.

### Author contributions

Benjamin R Thomson, Conceptualization, Data curation, Formal analysis, Validation, Investigation, Visualization, Methodology, Writing—original draft, Project administration, Writing—review and editing; Isabel A Carota, Conceptualization, Investigation, Methodology; Tomokazu Souma, Conceptualization, Writing—review and editing; Saily Soman, Formal analysis, Investigation, Writing—review and editing; Dietmar Vestweber, Resources, Writing—review and editing; Susan E Quaggin, Conceptualization, Formal analysis, Supervision, Funding acquisition, Validation, Project administration, Writing—review and editing

### Author ORCIDs

Benjamin R Thomson [iD] https://orcid.org/0000-0001-6565-5866
Isabel A Carota [iD] https://orcid.org/0000-0002-7980-2377
Tomokazu Souma [iD] https://orcid.org/0000-0002-3285-8613
Dietmar Vestweber [iD] http://orcid.org/0000-0002-3517-732X
Susan E Quaggin [iD] https://orcid.org/0000-0002-3706-5727

### Ethics

Animal experimentation: This study was performed in strict accordance with the recommendations in the Guide for the Care and Use of Laboratory Animals of the National Institutes of Health and the ARVO guidelines for care and use of vertebrate research subjects in eye research. All animal experiments were approved by the Animal Care Committee at the Center for Comparative Medicine of Northwestern University (Evanston, Illinois, USA) under animal protocols IS00002777, IS00006571 and IS00003091.

### Decision letter and Author response

Decision letter https://doi.org/10.7554/eLife.48474.SA1
Author response https://doi.org/10.7554/eLife.48474.SA2

## Additional files

### Supplementary files

• Transparent reporting form

## Data availability

All data described have been included in the manuscript. No data sets were generated during the course of this study.

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
