## [Decision Letter]

Thank you for submitting your article "Targeting the phosphatase PTPRB protects against retinal ganglion cell loss in a pre-clinical model of glaucoma" for consideration by *eLife*. Your article has been reviewed by three peer reviewers, one of whom is a member of our Board of Reviewing Editors, and the evaluation has been overseen by Didier Stainier as the Senior Editor. The reviewers have opted to remain anonymous.

The reviewers have discussed the reviews with one another and the Reviewing Editor has drafted this decision to help you prepare a revised submission.

Summary:

The authors present the clinical significance of their work up-front: glaucoma is a major cause of blindness worldwide and elevated intraocular pressure (IOP) from decreased aqueous humor outflow via inadequate Schlemm's canal (SC) integrity is a major risk factor for this event. Thomson et al. simply and elegantly demonstrate that haploinsufficiency of VEPTP results in upregulation of Tie2 signaling in vivo and rescues SC agenesis resulting from Tie2 haploinsufficiency. They further demonstrate that compound heterozygosity of VEPTP/*Tek* rescues elevated the IOP and retinal ganglion cell loss observed in *Tek^+/-^* mice. The authors acknowledge the extensive prior work performed by their group and others in this area. The current work adds to this literature by confirming the gene dosage effect of *Tek* and *Ptprb* in maintenance of Schlemm's canal and supports the further development of VEPTP inhibitors in the treatment of elevated IOP. The experimental approach is straightforward and logical. Figures are easy to interpret, and the conclusions are well-supported.

Essential revisions:

1) Introduction/Discussion: Clearly state that studied phenotypes are developmental rather than acquired. Alternatively, consider adding IOP studies with VE-PTP inhibitor in the *Tek^+/-^* mouse.

2) Figure 1: Include pTyr signaling in the *Ptprb;Tek* compound heterozygotes. Normalization of pTyr relative to *Tek^+/-^* alone would support the conclusion that upregulation of Tie2 signaling is the mechanism of SC/IOP rescue. This could be demonstrated in the eye by western and/or staining.

3) Figure 2: it would be useful to perform *Tek*, pTek, Prox1, and possibly Ki67 staining in the SC whole mounts, as the authors have done in previous studies. This would support the conclusion that the relative decrease in SC area is associated with defects in *Tek* signaling and subsequent Prox1 activation, as would be expected by the current model of SC development and maintenance. Given the ~two-fold variability in SC area across genotypes, a quantitative correlation may emerge. Short of this, the variability should be discussed.

4) The authors should show the retinal vascular phenotypes of the *Tek^+/-^, Ptprb*^NLS-LacZ/WT^ and compound *Tek^+/-^;Ptprb*^NLS-LacZ/WT^ mice in Figure 2.

5) Figure 3: Is there a retinal vascular phenotype in the *Ptprb^+/-^, Tek^+/-^* and compound heterozygote models? This could influence retinal ganglion cell counts.

6) The authors should make it clear to the readers that the model they use is a constitutive deletion, thus they describe developmental models rather than conditional gene deletion models. The latter would better mimic a therapeutic model.

---

## [Author Response]

Essential revisions:1) Introduction/Discussion: Clearly state that studied phenotypes are developmental rather than acquired. Alternatively, consider adding IOP studies with VE-PTP inhibitor in the Tek^+/-^ mouse.

Thank you for pointing out where our Discussion was unclear. We have emphasized this important point in the revised manuscript:

Abstract: “Here, we describe a novel approach to enhance canal formation in these animals by deleting a single allele of the gene encoding the phosphatase PTPRB during development.”

Introduction: “Here, we show that developmental deletion of a single *Ptprb* allele…”

Results: “Developmental deletion of a single *Ptprb* allele in *Tek^+/-^;Ptprb*^NLS-LacZ/WT^ mice…”

Discussion: “These defects were blunted by developmental *Ptprb* haploinsufficiency in *Tek*^+/-^;*Ptprb*^NLS-LacZ/WT^ mice…”

2) Figure 1: Include pTyr signaling in the Ptprb;Tek compound heterozygotes. Normalization of pTyr relative to Tek^+/-^ alone would support the conclusion that upregulation of Tie2 signaling is the mechanism of SC/IOP rescue. This could be demonstrated in the eye by western and/or staining.

We have attempted to quantify levels of *Tek* phosphorylation in Schlemm’s canal endothelium of haploinsufficient mice, but the low levels of total *Tek* in these animals has made it difficult to obtain convincing results. Quantification of immunostaining using phospho-TEK antibodies showed high variability, making it difficult to draw meaningful conclusions. Western blot of eye and lung tissue was similarly complicated by the lower overall level of TEK expression in haploinsufficient animals.

In light of this difficulty, we have expanded our Discussion to emphasize that while TEK is the most likely substrate, other targets of PTPRB may be implicated in its role in Schlemm’s canal.

3) Figure 2: it would be useful to perform Tek, pTek, Prox1, and possibly Ki67 staining in the SC whole mounts, as the authors have done in previous studies. This would support the conclusion that the relative decrease in SC area is associated with defects in Tek signaling and subsequent Prox1 activation, as would be expected by the current model of SC development and maintenance. Given the ~two-fold variability in SC area across genotypes, a quantitative correlation may emerge. Short of this, the variability should be discussed.

We thank the reviewers for these useful suggestions. We have performed whole mount staining of SC at P5 (during canal development) and at P20 (after development is largely complete). These experiments revealed reduced proliferation of PROX1-positive SC endothelial cells (Ki67/PROX1 double positive cells) in *Tek* haploinsufficient mice compared to control littermates at P5, possibly explaining the smaller Schlemm’s canal observed in these animals. In contrast, *Ptprb/Tek* double heterozygous animals exhibited a normal proportion of Ki67 positive cells, consistent with their increased SC area. This new data has been included in the manuscript as Figure 2B. Likewise, and consistent with previous work from our lab and other groups (Kim et al., 2017), PROX1 expression is reduced in SC of *Tek*^+/-^ eyes. This reduced PROX1 expression is absent in *Ptprb/Tek* double heterozygous animals, which have normal expression. This data has been included in the manuscript as new Figure 2D.

The reviewers raised an important point regarding the variability in SC area, especially as seen in *Ptprb/Tek* double heterozygous animals. We have previously reported that *Tek^+/-^* mice exhibit a dramatic increase in Schlemm’s canal focal malformations, convolutions and narrowings compared to littermate controls. To address the reviewer’s question, we now include data which clearly shows that while Schlemm’s canal area is substantially larger in *Ptprb/Tek* double heterozygous animals when compared to *Tek^+/-^* littermates, this overall morphological phenotype remains—suggesting that ANGPT-TEK signaling may have a guiding role in canal development which is not recapitulated by nonspecific receptor activation. We have included this data in the manuscript as a new Figure 2E.

4) The authors should show the retinal vascular phenotypes of the Tek^+/-^, Ptprb^NLS-LacZ/WT^ and compound Tek^+/-^;Ptprb^NLS-LacZ/WT^ mice in Figure 2.

We thank the reviewer for raising this important point. We have examined retinal vasculature in littermate mice of all genotypes at P5 (during retinal angiogenesis) and at P20 (after the vascular front has reached the retinal periphery) and found no difference in retinal vascularization. By P20, normal patterning was observed in all three vascular layers and hyaloid vessels were absent in all animals. These new data has been included in the revised manuscript as Figure 3—figure supplement 1.

5) Figure 3: Is there a retinal vascular phenotype in the Ptprb^+/-^, Tek^+/-^ and compound heterozygote models? This could influence retinal ganglion cell counts.

Please see our response to question #4 (above). Retinal vasculature appeared normal in all animals examined and we have included these data in the revised manuscript.

6) The authors should make it clear to the readers that the model they use is a constitutive deletion, thus they describe developmental models rather than conditional gene deletion models. The latter would better mimic a therapeutic model.

This is a key point in our manuscript, and we appreciate the feedback that it should be clearer. We have emphasized this point in the revised paper as described in our response to point 1 (above).